# 3D Method for Occlusal Tooth Wear Assessment in Presence of Substantial Changes on Other Tooth Surfaces

**DOI:** 10.3390/jcm9123937

**Published:** 2020-12-04

**Authors:** Nikolaos Gkantidis, Konstantinos Dritsas, Christos Katsaros, Demetrios Halazonetis, Yijin Ren

**Affiliations:** 1Department of Orthodontics and Dentofacial Orthopedics, School of Dental Medicine, University of Bern, CH-3010 Bern, Switzerland; konstantinos.dritsas@zmk.unibe.ch (K.D.); christos.katsaros@zmk.unibe.ch (C.K.); 2Department of Orthodontics, W.J. Kolff Institute, University Medical Center Groningen, University of Groningen, 9700RB Groningen, The Netherlands; y.ren@umcg.nl; 3Department of Orthodontics, School of Dentistry, National and Kapodistrian University of Athens, GR-11527 Athens, Greece; dhalaz@dent.uoa.gr

**Keywords:** tooth wear, measurement method, quantitative assessment, three-dimensional imaging, surface model, three-dimensional superimposition, orthodontic retention, fixed retainers

## Abstract

Early diagnosis and timely management of tooth or dental material wear is imperative to avoid extensive restorations. Previous studies suggested different methods for tooth wear assessment, but no study has developed a three-dimensional (3D) superimposition technique applicable in cases where tooth surfaces, other than the occlusal, undergo extensive morphological changes. Here, we manually grinded plaster incisors and canines to simulate occlusal tooth wear of varying severity in teeth that received a wire retainer bonded on their lingual surfaces, during the assessment period. The corresponding dental casts were scanned using a surface scanner. The modified tooth crowns were best-fit approximated to the original crowns using seven 3D superimposition techniques (two reference areas with varying settings) and the gold standard technique (GS: intact adjacent teeth and alveolar processes as superimposition reference), which provided the true value. Only a specific technique (complete crown with 20% estimated overlap of meshes), which is applicable in actual clinical data, showed perfect agreement with the GS technique in all cases (median difference: −0.002, max absolute difference: 0.178 mm^3^). The outcomes of the suggested and the GS technique were highly reproducible (max difference < 0.040 mm^3^). The presented technique offers low cost, convenient, accurate, and risk-free tooth wear assessment.

## 1. Introduction

Tooth wear characterizes the superficial loss of tooth matter over time. In humans, tooth wear occurs as a consequence of normal function, parafunction, or environmental factors, such as very acidic food. The main issue of concern, when excessive tooth wear occurs, is impaired dental esthetics. However, it may also affect facial morphology and speech, impacting patients’ quality of life. In recent years, the need for retaining natural teeth intact for several decades has arisen due to the increase in life expectancy. This, along with increased patient esthetic demands has designated tooth wear as an important problem that needs to be addressed [1]. Advances in the dental field, including tooth wear management, enabled teeth maintenance till late stages of life [2].

Early diagnosis is imperative for the timely management of tooth wear, to avoid subsequent problems and extensive restorations [2,3]. To facilitate tooth wear assessment, several previous studies have suggested a variety of qualitative and quantitative methods. Qualitative approaches are subjective and usually suffer from reduced precision and reproducibility [4,5,6]. On the contrary, certain quantitative techniques have shown adequate reproducibility, usually within the range of 15–20 μm, concerning vertical loss of tooth structure. However, the trueness of these techniques remains questionable, since studies lack a gold standard reference for comparison [7]. Other shortcomings include the high complexity of these techniques that need special equipment and expertise to be applied properly, as well as the dental impressions and physical models required. The above increase the costs, the inaccuracies, and the applicability of the techniques in actual clinical conditions [7,8,9].

A highly accurate three-dimensional (3D) superimposition technique has been suggested recently for occlusal tooth wear assessment using serial digital dental models [10]. Assuming that relevant software and hardware is accessible, which is realistic for a contemporary practice, this technique is applicable under regular clinical conditions. The existence of a previous dental model, obtained at a certain point in the past, is a prerequisite to apply this technique. The rapid incorporation of intraoral scanners in contemporary dentistry [11] and the use of relevant software for various applications facilitates this purpose. The aforementioned previous study [10] focused on occlusal tooth wear assessment in cases that only the occlusal tooth surfaces were subjected to changes over time, offering an accuracy of 0.033 mm^3^, which corresponds to approximately 9 μm of vertical tooth loss [10]. This accuracy level is much higher than any level than might be considered clinically significant. However, other tooth surfaces might also be affected by function, pathology, or due to iatrogenic interventions [12,13,14]. This might complicate occlusal tooth wear assessment through serial 3D surface model superimpositions. To our knowledge, there is no study in the literature that has addressed this issue. A common instance where tooth surfaces, other than the occlusal, undergo extensive morphological changes regards the placement of bonded wire retainers. Such retainers might be bonded following orthodontic treatment, to stabilize teeth after trauma, or in periodontally compromised patients [14,15,16,17]. Thus, the purpose of the present study was to develop and validate a 3D occlusal tooth wear assessment technique, applicable when the anterior teeth crown morphology is highly altered during the testing period, in non-occlusal surfaces.

## 2. Experimental Section

### 2.1. Ethical Approval

The research project is registered and approved by the Swiss Ethical Committee of the Canton of Bern (Protocol No. 2019-00326). All experiments were performed under the relevant regulations and according to the pre-specified protocol. All participants signed an informed consent prior to the use of their data in the study.

### 2.2. Sample

The present sample was derived from an existing material, which was previously generated and thoroughly described [10]. In this study, sixteen dental plaster models (type IV plaster, white color, Fujirock EP Premium, GC, Leuven, Belgium) with (*n* = 8; 4 maxillary and 4 mandibular; crowding ≤ 1 mm) and without (*n* = 8; 4 maxillary and 4 mandibular; crowding: 4–10 mm) well aligned dental arches were used. These were retrieved from the archive of the Department of Orthodontics and Dentofacial Orthopedics, University of Bern, Switzerland. The models represented individuals with natural permanent dentition and no extreme morphological variation in oral structures (visual inspection).

### 2.3. Tooth Wear Simulation

According to a pre-specified and previously validated protocol, eighteen canines and eighteen incisors, equally distributed among the dental models described above, were manually grinded to simulate occlusal tooth wear of varying severity (approximately 0.5, 1, and 2 mm of vertical loss, respectively) [10]. Grinding was performed both symmetrically and asymmetrically, using a dental laboratory straight handpiece or a dental laboratory stone knife, to simulate a variety of normal tooth wear patterns. Additionally, for the needs of the present study, a fixed retainer that is usually bonded following orthodontic treatment or to stabilize highly mobile teeth following trauma or severe periodontally compromised teeth, was simulated in the anterior teeth. For this, a twisted white coated ligature wire (0.3 mm initial diameter, Dentaurum, Ispringen, Germany) was placed on the middle of the lingual surfaces of the test anterior teeth. A white modeling compound (Play-Doh putty, Hasbro, Pawtucket, RI, USA) was used to stabilize the wire on the teeth during scanning and simulate the bonding material placed in vivo. Both materials were selected after pilot testing that confirmed sufficient material surface acquisition by the scanner. The setting resulted in two intact teeth adjacent to each grinded tooth that received the retainer. The intact teeth and other adjacent anatomical structures that were not artificially altered, were used as superimposition reference areas to provide the gold standard (true) measurement [10,18].

### 2.4. 3D Model Acquisition

The dental casts of the before (T0) and after wear simulation plus retainer placement (T1) conditions, were scanned using a laboratory 3D surface scanner (stripe light/LED illumination; full dental arch accuracy <20 μm; Laboratory scanner D104a, Cendres + Métaux SA, Biel/Bienne, Switzerland). Repeated single jaw model scans with this scanner, show a distance between corresponding surfaces always smaller than 5 μm. The subsequent binary 3D Standard Tessellation Language (STL) models were imported in Viewbox 4 software (version 4.1.0.1 BETA, dHAL Software, Kifissia, Greece) to apply the methods tested in the study. Each such maxillary or mandibular full dental arch model consisted of 600.000–900.000 triangles.

### 2.5. Tooth Wear Volume Measurement Workflow

Following 3D superimpositions through seven test techniques and the gold standard technique, the crowns of the grinded teeth (T1) were manually segmented and compared to the segmented original crowns (T0).

For each tested tooth, the gold standard (GS) measurements were obtained through T0/T1 model superimpositions on intact adjacent teeth and alveolar processes (Figure 1a). Perfect matching is expected in these areas following a best-fit superimposition, which enables accurate occlusal tooth wear assessment [10,11,18].

Measurements were performed according to a modified, previously published protocol [10]. The first group of measurements (PC: partial crown) was obtained using part of the T0 clinical crown as superimposition reference (Figure 1b–d). This aimed to include crown areas that could be considered unaffected from occlusal wear or retainer placement. On the other hand, the complete T0 clinical crown was used as a superimposition reference for the second group (CC: complete crown) (Figure 1e). Each time, the T0/T1 3D models of each patient were superimposed using the software’s implementation of the iterative closest point algorithm (ICP) [19], with predefined settings that are described below. The whole process always started from the original initial position of the models. Thus, the first step included the manual approximation of the two objects to facilitate rapid automatic registration through the ICP algorithm. For the same reason, following the manual approximation, in cases of a setting that included less than 50% estimated overlap of meshes, a partial approximation of the two models was always performed, using 100% estimated overlap of meshes, before applying the predefined setting.

Based on previous experience [10] and pilot testing, the performance of different ICP settings was assessed using the superimposition reference areas described above (Figure 1). The eight specific techniques applied in the study (combinations of ICP settings and reference areas) are listed in Table 1. The basic setting consisted of 100% estimated overlap of meshes, matching point to plane, exact nearest neighbor search, 100% point sampling, exclude overhangs, and 50 iterations. Other ICP settings were identical to the basic setting, but with user defined, 40%, or 20% estimated overlap of meshes. The user defined estimated overlap of meshes was freely chosen by the operator for each individual measurement, through an iterative process. Each time, following various adjustments of the specific setting, the user aimed to achieve the maximum overlap of the superimposed teeth and, primarily, of the adjacent intact structures. The selected value that provided the best overlap of intact structures in each case, was noted in an Excel sheet. For each reference area, the average of these values, provided the additional estimated overlap settings to be tested. It should be noted that the case-specific, user defined setting is impossible to be obtained in actual patient data, since there are no absolutely intact structures in the mouth between two time points [11]. Thus, this approach allowed the determination of a setting that might work properly, while being applicable in actual clinical conditions.

The superimposed 3D tooth crown models through each technique described above, were simultaneously sliced using one (gingival) to three planes (gingival, mesial, and distal), based on a previously published protocol [10]. For each pair of teeth, the slicing planes were positioned to include the complete occlusal wear surface. At the same time care was taken to avoid significant differences between the two models at the edges of the crown parts to be sliced. This was verified though the visualization of relevant color coded distance maps. Consequently, identical filling of the holes of each occlusal part of the sliced T0 and T1 crown models was achieved (Figure 2). In certain cases, the hole created from slicing had to be split to two or more parts, through manual connection of contralateral points in sharp edges. This ensured that the edges of each hole to be filled, through the application of the software’s algorithm, were located on the same plane. Thus, it ensured identical hole filling and creation of watertight T0 and T1 3D models that were matching at all parts, apart from the occlusal part that was the one to be assessed (Figure 3). Following this approach, occlusal tooth wear was defined as the difference between these two superimposed crown parts. The wear amount, expressed as volume loss of tooth structure (mm^3^), that was detected through the gold standard technique, was then compared to that of the test techniques.

To assess the error of Viewbox 4 software on volumetric assessments, the volumes of ten tooth parts similar to those used for the study were measured using Viewbox 4, Artec Studio 12 Professional (Artec 3D, Luxembourg), and MeshLab 2016.12 [20] and compared. The error attributed to the used surface scanner was measured in ten teeth of various types, located on ten different models that were scanned twice. Following superimposition of the corresponding identical teeth derived from repeated scans, volumes similar to those tested in the study were calculated. Zero difference in corresponding volumes would indicate perfect superimposition of the repeated models and zero scanner error. Reproducibility of tooth wear assessment techniques was tested through repeated measurements of ten teeth, on four randomly selected models, two maxillary and two mandibular (one with and one without crowding each) by one operator, following a 1-month period.

### 2.6. Statistical Analysis

The statistical approach followed here is comparable to that of a previous similar study [10]. Statistical analysis was carried out by using the IBM SPSS statistics for Windows (Version 26.0. IBM Corp: Armonk, NY, USA). Non-parametric statistics were applied based on abnormal distribution of the raw data of certain variables (Kolmogorov–Smirnov and Shapiro–Wilk tests).

Agreement between different techniques with the gold standard technique (trueness) regarding tooth wear, was shown in box plots. Perfect trueness is indicated by zero median value, whereas the larger the deviation from zero the lower the trueness. Within techniques, the range of deviation of individual values from the median value shows precision. Differences in trueness and precision among different techniques were tested using Friedman’s test, followed, where applicable, by Wilcoxon’s signed rank test for pairwise comparisons [10].

Potential effects of presence of crowding, tooth type, or tooth wear amount on the trueness and precision of each technique were explored through visual inspection of relevant plots and unpaired comparative tests within techniques [10].

Intra-operator error (reproducibility) of the gold standard and the technique of choice was assessed through Bland Altman plots, with markers set by tooth category. Any deviation from zero shows imprecision of the technique. Differences in the reproducibility of the two techniques were tested in an unpaired manner through Mann–Whitney U test [10].

For all tests, a two-sided significance test was carried out at an alpha level of 0.05. When multiple pairwise comparisons were performed, the level of significance was altered according to the Bonferroni adjustment.

## 3. Results

Regarding the volumetric assessment of tooth crown parts of interest, all three tested software provided identical values for all tested models. Following superimposition of corresponding single teeth, derived from repeated scans, with the CC(C) technique, the median volume difference was 0.0115 mm^3^ (range: 0.0003, 0.0350 mm^3^) when all differences were transformed to absolute values. The original position of the second scan was randomly altered prior to these tests. This value indicates the scanner plus the superimposition error regarding tooth wear assessment using single teeth. Repeated tooth wear measurements with the gold standard technique and the CC(C) technique of choice showed no systematic error (one sample *t*-test, *p* > 0.05). The amount of tooth wear and the tooth type did not seem to affect reproducibility. The outcomes of both techniques were considered highly reproducible overall and in individual measurements (max difference < 0.040 mm^3^) (Figure 4). There was no difference in the reproducibility of the gold standard and the technique of choice (*p* = 0.94). The superimposition error of the gold standard technique that provided the reference measurements is shown in Figure 4a and it was on average less than 0.005 mm^3^ and in all cases less than 0.020 mm^3^.

There were significant differences in the trueness of the tested techniques (Friedman test: *p* < 0.001). PC(A) and PC(B) differed clearly from all other techniques. Among the rest, all techniques differed significantly to each other apart from CC(A) vs. PC(D), CC(B) vs. PC(C), CC(B) vs. CC(C), and CC(C) vs. PC(C) (Wilcoxon signed rank test: *p* < 0.01; Figure 5). Analogous differences in precision were evident among all techniques (Figure 5). The CC(C) technique was the only technique that showed perfect agreement with the GS technique in all cases (median difference: −0.002, max absolute difference: 0.178 mm^3^) and is applicable in actual clinical conditions.

Tooth type did not affect the trueness of each technique (Mann–Whitney U test, *p* > 0.05). However, measurements in canines tended to be more precise compared to those of incisors (Figure 6a). Tooth alignment in the dental arches also did not affect the outcomes of any technique (Mann–Whitney U test: *p* > 0.05). Tooth wear amount did not show any significant effects (Kruskal–Wallis test, *p* > 0.01), though there was a tendency for slightly reduced trueness and precision in the group with the highest amount of tooth wear (Figure 6b).

The difference between the CC(C) technique of choice and the GS technique was always small for any tested tooth type or amount of tooth wear (Appendix A). However, there was a tendency for decreased precision in case of 2 mm vertical height loss of tooth structure (Kruskal–Wallis test, *p* = 0.029). Pairwise comparisons showed a significant difference only between the 0.5 mm and the 2 mm groups (Mann–Whitney U test, *p* < 0.01).

## 4. Discussion

Additional to proper diagnosis, easily applicable methods that facilitate accurate assessment of tooth wear will support the scientific community in better understanding potential causes, acting mechanisms, and contributing factors. Material wear is also very important for materials science when conducting in vitro tests, but also to clinically test the real-life performance of materials [21,22]. Here we present a highly accurate and informative tooth wear assessment method, which is also more convenient than the already available laboratory methods. It allows the measurement and visualization of occlusal tooth wear in three dimensions of space, on patients that received a bonded wire retainer during the assessment period. To our knowledge, this is the first study that addresses this issue. The suggested method requires the existence of two serial dental models to assess wear progress over time. In the era of digitization, this is not considered a limitation, since it is expected that soon an intraoral patient scan will be an integral part of basic dental diagnostic records. Providing that the suggested superimposition method registers sufficiently serial surface models similar to those tested here, it can be used for any relevant purpose, including occlusal material wear testing.

The present method offers the opportunity for detailed assessment of changes over time, in corresponding surfaces, without the need for placing any landmarks, which might be a time-consuming and error prone process [23,24]. Apart from the clinically relevant applications, following the proper superimposition of serial 3D tooth models, which enables the accurate detection of the worn tooth part during a specific time period, this method can be combined with various other types of 3D shape analysis, such as those used for ecometrics in dental ecology [25], including dental topography methods [24]. The method works on a μm scale that is defined by its accuracy, as well as by the resolution of the applied scanning systems. For example, based on scanner resolution of the intraoral, as well as the current dental lab scanners [11], microwear analysis might not be possible, though it might be quite useful for testing certain hypotheses [26].

In a previous study we developed a similar method [10] applicable on teeth that did not undergo significant alterations in any crown surface, apart from the occlusal. Epidemiologic studies indicate that in modern humans, tooth wear, especially of the clinical crown, mainly concerns the occlusal part of the teeth due to the direct contacts with the antagonists [12,13]. Thus, the simulation model that was previously developed and applied here, represented adequately the most common clinical occurrence. However, certain patients may receive a fixed retainer at their buccal or more often at their lingual surfaces. This usually consists of a continuous wire bonded with composite resin at the middle of the respective tooth surfaces. This type of retainer is common in patients that received orthodontic treatment, but also in severe periodontally compromised patients, or in patients that required tooth stabilization following dental trauma [14,16,17]. Thus, we suspected that our previously suggested method might not perform similarly when a fixed retainer has been bonded to the test teeth during the assessed period. Indeed, after testing various possibilities for the latter case, we concluded that the complete crown with reduced estimated overlap of meshes to 20%, namely the CC(C) technique, provided the most accurate outcomes. The previous study, on intact teeth in non-occlusal surfaces, suggested an estimated overlap of meshes of 30% for optimal outcomes [10].

The present method offers a median accuracy of 0.002 mm^3^, with the worst individual measurement showing a difference of 0.178 mm^3^ from the gold standard measurement. The gold standard measurement always showed a precision higher than 0.020 mm^3^. The scanner plus the superimposition error for single teeth measurements, using the suggested technique, was always smaller than 0.035 mm^3^. Thus, even in the unfavorable scenario of extensive changes in non-occlusal surfaces, this method performs quite satisfactorily. A freeware and easy to use software (WearCompare, School of Dentistry, University of Leeds, Leeds, UK) [27] providing tooth wear measurements, following serial tooth surface model superimpositions, showed high performance when identical (duplicated) surfaces were used as superimposition references. However, when the original position of the duplicated models was altered, the software showed errors in volumetric assessment as high as 1 mm^3^. Furthermore, when actual patient data were considered, it showed considerable and statistically significant differences on volumetric assessments from Geomagic Control software (3D Systems, Darmstadt, Germany), averaging 0.59 mm^3^ [27]. This software uses the buccal and lingual surfaces as superimposition references to assess occlusal tooth wear. We also tested this approach both in a previous [10] and in the current study and it did not provide superior outcomes to that of the suggested techniques, which use the whole crown as superimposition reference. Furthermore, the performance of WearCompare software in presence of considerable changes in the used superimposition reference areas remains to be tested.

The current settings might also be applicable to patients that show extensive tooth wear on buccal or lingual surfaces [12,15]. In such cases, apart from the occlusal surface, another surface is highly altered over time, such as with the placement of a retainer. Following the superimposition of serial tooth models, the operator can generate color coded distance maps and confirm the applicability of the suggested technique, if the detected changes are comparable to those simulated here. For example, in cases of erosion, changes might be present on the entire tooth surface and, thus, the suggested technique might not be applicable. Finally, if the teeth are not subjected to any change between two time points, the present technique still performs a complete registration of the two surface models (Appendix A). However, this will be a rare occasion, since in living humans, minimal changes in form or in spatial relations are always expected over time. Even if there is no morphological change, two serial intraoral scans will not provide identical models [11].

In actual patients, the gingival margin area, and thus the corresponding clinical crown structures, might also change over time. However, with the present method this is not a limitation, since in case of large differences in the gingival part of the crown, the operator can select the shortest clinical crown as the superimposition reference. Then, by selecting the “exclude overhangs” option, which is included in the suggested default settings, the software ignores the part of crown that does not exist in both models. The performance of this function has been extensively applied and tested in previous studies, on various types of surface models and has always provided solid outcomes [10,11,18,28,29,30].

The present study simulated tooth wear amounts of approximately 0.5, 1, and 2 mm to represent variations of clinically significant loss of tooth crown parts, in terms of dental appearance. As evident in an analogous study [10], the high performance of the technique presented here was not considerably affected by tooth wear amount. Smaller amounts of occlusal tooth wear might be present in actual patients, but were not simulated here. We considered the tested range more relevant, since the ICP algorithm would perform equally well, if not better, in cases of smaller surface changes (Appendix A) [11,18,28]. Providing adequate superimposition of serial tooth models under the tested circumstances, the accuracy of the suggested method is defined by its difference from the tooth wear value obtained using the gold standard method. The amount of tooth wear that can be safely detected by the method is determined by the specified accuracy level. However, a potential improvement of the accuracy level in cases of minimal tooth wear remains to be verified by future research.

The high accuracy laboratory scanner used here to generate the surface models, has comparable accuracy to the current high end intraoral scanners, considering relatively small structures, such as single tooth crowns [11,18,31]. In the present study, we selected this scanner to eliminate the scanner error effect on the gold standard measurement, which requires a larger superimposition reference area. This measurement is feasible only with the present experimental design, where the structures adjacent to a worn tooth are intact. The true wear amount is measured through the use of these intact structures as superimposition reference. This measurement is then compared to the measurement obtained using the single tooth crown as superimposition reference. Only the latter option is feasible in actual clinical conditions, where no structure adjacent to single teeth can be considered morphologically stable over time. This is the main reason why gold standard measurements (true values) cannot be available when using actual clinical data.

So far, most methods used to measure tooth wear are based on qualitative assessments with limited precision or on complex laboratory quantitative approaches that require expertise and special equipment to be applied [7]. The presented method requires 3D tooth surface models that can nowadays be easily obtained through intraoral scanners [11]. These models can then be processed using software applications that are widely available under reasonable costs [32]. Thus, following the necessary short-term training, these methods can be potentially incorporated in a regular dental practice environment, enhancing the diagnostic ability of occlusal tooth wear, especially in cases that this is considered critical. Apart from this major advantage, the performance of this method is also comparable to, if not higher than most available methods [7]. So far, a direct comparison with other methods is not possible, since no other study has performed quantitative tooth wear assessment in teeth that underwent significant changes in surfaces other than the one tested. Among others, a significant strength of the present in vitro study is that it allowed the comparison of different techniques with a gold standard technique, which provided the true measurement. Thus, apart from reproducibility, the study tested the trueness of the measurements, which is usually a missing part from previous reports [7]. Furthermore, the present method is superior to methods that perform 2D measurements, such as the vertical loss of tooth structure, since it offers much higher amount of 3D information. In addition to the quantification of tooth volume loss, this 3D superimposition technique enables the visualization of a color coded distance map that allows a thorough quantitative and qualitative assessment of tooth wear in the 3D space. Thus, following the proper application of the suggested technique, patterns of tooth wear or any spatial differences in the tooth surfaces, can be easily identified and quantified according to the needs of each individual test.

Our research group has worked extensively in testing surface-based 3D superimposition in the craniofacial area, showing promising results in many different applications [10,11,18,28,29,30,33]. In the aforementioned studies, the performance of the used software and the specific algorithm has been thoroughly tested and shown highly reproducible results. Thus, the high reproducibility of the present techniques was an expected finding. Furthermore, the technique of choice consistently provided values very close to the true amount of tooth wear. The individual differences of the selected CC(C) technique from the true value were almost always smaller than 0.1 mm^3^. This accuracy level is comparable to that achieved on teeth that underwent changes only at their occlusal surfaces [10]. However, the current approach consists a clinically applicable technique in a more challenging scenario, where the used tooth surfaces have been considerably changed over time.

In diagnostic accuracy studies, it is of utmost importance to assess differences between techniques or repeated measurements in each individual case and not only between group means [34]. In the latter, positive and negative differences between diverse cases can be eliminated providing the misleading impression of similar outcomes. However, it is important to have accurate measurements in each individual case, since, for example, this information might affect treatment decisions. Indeed, in the present study the mean values of all techniques were very close to each other and to the true value. However, when results of individual cases were considered, the superiority of the selected CC(C) technique became evident.

A limitation of the present study could be that only anterior teeth were thoroughly tested. This approach was followed because these teeth usually require a fixed retainer [14,16,17]. Furthermore, we also included canines in our tests, which are more round-formed teeth. Along with the present results, previous results have also shown that tooth type does not considerably affect the outcomes [10]. Thus, we expect that this technique will be also applicable on posterior teeth (Appendix A). Another limitation could be that intra-operator error was only assessed for the gold standard and the technique of choice. This was considered adequate since the two techniques showed satisfactory agreement and both provided excellent reproducibility outcomes. Furthermore, as described above, thorough error evaluation of similar techniques has been published previously and always provided favorable outcomes [10].

## 5. Conclusions

The present report suggests a 3D superimposition technique to assess occlusal tooth wear in anterior teeth that received a bonded wire retainer during the assessment period. Following the superimposition of serial tooth crown models, loss of tooth structure can be visualized and quantified in all dimensions of space. The technique is highly accurate, informative, and relatively easy to use, and thus, it will facilitate associated research, but it can also be easily incorporated in a clinical environment. Further work is required to verify the performance of the technique on posterior teeth and regarding smaller increments of tooth wear.

## Figures and Tables

**Figure 1 jcm-09-03937-f001:**
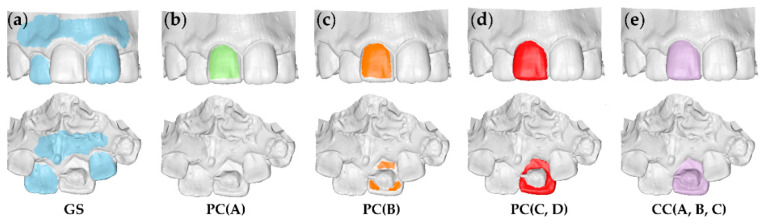
Reference areas used to measure tooth wear at the right maxillary permanent central incisor. (**a**) Gold standard area (GS: blue). (**b**–**d**) Partial crown areas (PC: green, orange, red). (**e**) Complete crown area (CC: purple). The upper row shows the buccal and the lower row the palatal view. The letters within the parentheses indicate the settings applied for each reference area.

**Figure 2 jcm-09-03937-f002:**
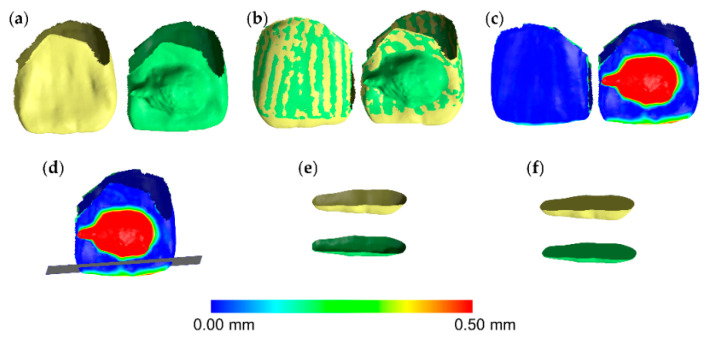
Tooth wear measurement process in a maxillary central incisor. (**a**) Tooth before (yellow) and after (green) tooth wear and retainer simulation. (**b**) Superimposed tooth crowns using the complete crown technique and setting C (20% estimated overlap) shown from the buccal (left) and the lingual (right) aspect. (**c**) Color coded distance map showing the tooth wear from the buccal (left) and the lingual (right) side. (**d**) Level (grey) used to simultaneously slice the two crowns. (**e**) Sliced tooth crowns. (**f**) Holes filled to create watertight models, and thus, calculate volumes.

**Figure 3 jcm-09-03937-f003:**
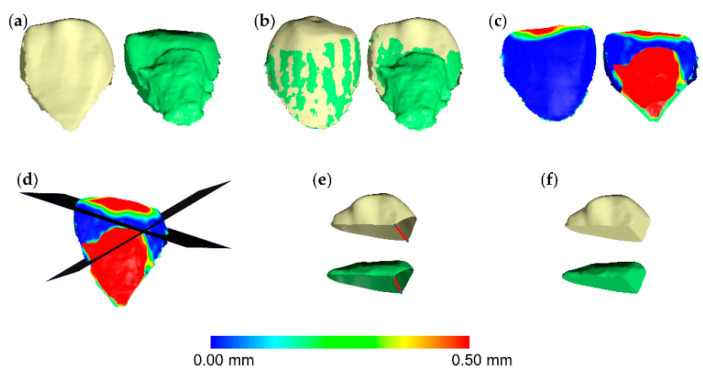
Tooth wear measurement process in a mandibular canine. (**a**) Tooth before (yellow) and after (green) tooth wear and retainer simulation. (**b**) Superimposed tooth crowns using the complete crown technique and setting C (20% estimated overlap) shown from the buccal (left) and the lingual (right) aspect. (**c**) Color coded distance map showing the tooth wear from the buccal (left) and the lingual (right) side. (**d**) Two levels (grey) used to simultaneously slice the two crowns. (**e**) Sliced tooth crowns with connected contralateral points (blue) in sharp edges (red line), splitting the hole in two parts to ensure identical hole filling process. (**f**) Holes filled to create watertight models, and thus, calculate volumes.

**Figure 4 jcm-09-03937-f004:**
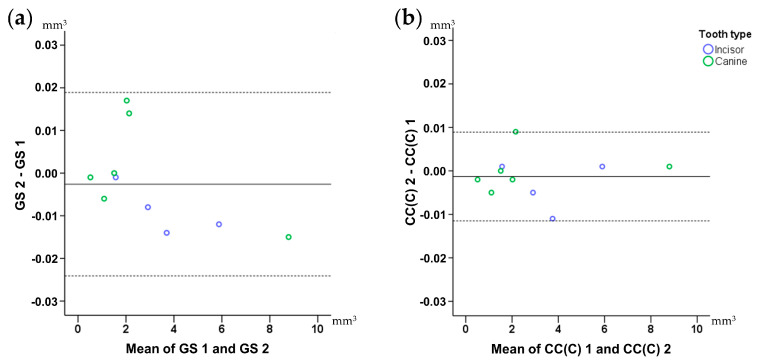
Bland–Altman plots showing intra-operator error on tooth wear volume measurements. (**a**) Gold standard (GS) technique. (**b**) Technique of choice: complete crown (CC) with 20% estimated overlap. The axes length represents the true range of measured tooth wear values. The continuous horizontal line shows the mean and the dashed lines the 95% confidence intervals.

**Figure 5 jcm-09-03937-f005:**
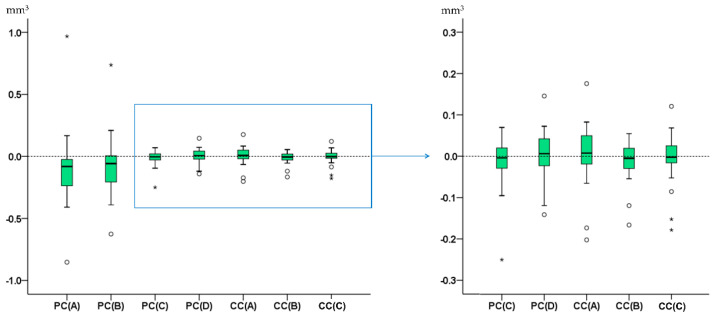
Box plots showing in the Y-axis the difference of each technique with the gold standard technique in tooth wear measurements. The upper limit of the black line represents the maximum value, the lower limit the minimum value, the box the interquartile range, and the horizontal black line the median value (trueness). Outliers are shown as black circles (°) or asterisks (*), in more extreme cases, with a step of 1.5 × IQR (interquartile range). Zero value (dashed horizontal line) indicates perfect agreement with the gold standard. The vertical length of each plot indicates precision. The blue box on the left image indicates the area of the graph shown in the right image in a larger scale. PC: partial crown; CC: complete crown.

**Figure 6 jcm-09-03937-f006:**
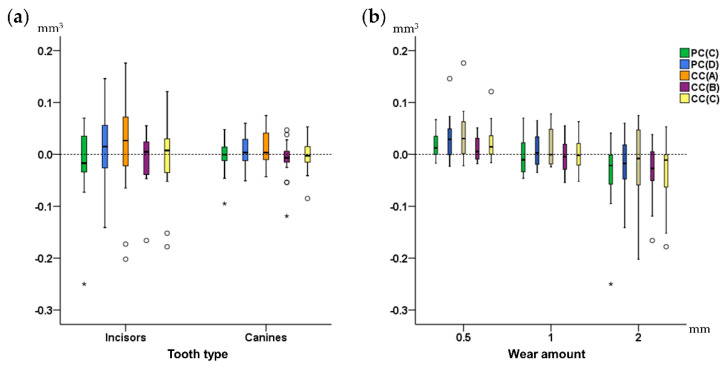
Box plots showing on the Y-axis the difference of each technique from the gold standard technique in tooth wear measurements, (**a**) by tooth type, and (**b**) by amount of tooth wear. The upper limit of the black line represents the maximum value, the lower limit the minimum value, the box the interquartile range, and the horizontal black line the median value (trueness). Outliers are shown as black circles or asterisks, in more extreme cases, with a step of 1.5 × IQR (interquartile range). Zero value (dashed horizontal line) indicates perfect agreement with the gold standard. The vertical length of each plot indicates precision. PC: partial crown; CC: complete crown.

**Table 1 jcm-09-03937-t001:** Superimposition techniques tested in the study.

Technique	Reference Area	Estimated Overlap
GS	Adjacent intact teeth and alveolar processes	100%
PC(A)	Buccal surface	100%
PC(B)	Buccolingual surfaces without composite	User defined
PC(C)	Complete crown without composite	User defined
PC(D)	Complete crown without composite	40%
CC(A)	Complete crown	40%
CC(B)	Complete crown	User defined
CC(C)	Complete crown	20%

GS: gold standard; PC: partial crown; CC: complete crown.

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
