# Peer review of "3D Method for Occlusal Tooth Wear Assessment in Presence of Substantial Changes on Other Tooth Surfaces"

_jcm, 2020, doi:10.3390/jcm9123937_

Round 1

Reviewer 1 Report

I enjoyed reading your paper and appreciate that you are bringing an insight to the wider dental audience, of the fact that computers cannot simply align STLs perfectly every time.

However, I think there are some significant issues and omissions that would require addressing in the methodology before this work could truly be said to be advancing the body of knowledge. I outline these below and hope you find this input useful in your future work.

The digital assessment of tooth wear is an active area of research, but the introduction misses much of the current research (which in fact, points towards there being a validated, free and simple to use tool – WearCompare – for exactly the purpose this paper investigates). See the following reference, for example.

O'Toole, S., Osnes, C., Bartlett, D., & Keeling, A. (2019). Investigation into the validity of WearCompare, a purpose-built software to quantify erosive tooth wear progression. Dental Materials, 35(10), 1408-1414. doi:10.1016/j.dental.2019.07.023 

In addition, the above method has been validated by specialists in metrology (rather than ‘just’ dentists with little formal background in digital/mathematical techniques). See link below for a recent paper.

https://eprints.soton.ac.uk/444540/

The introduction should include the state of the art, and your experiment should have compared the results from this method.

Method

The introduction talks about intraoral scanning (IOS), but your research uses extraoral (model) scanning which is known to be higher resolution. Therefore your results are only applicable to model scanning and you should not mention IOS, or make clear the shortcomings of the study in this regard.

There are some issues with the method meaning you cannot conclude ‘The presented technique offers low cost, convenient, accurate, and risk-free tooth wear assessment’ in your abstract.

You only test incisors and canines, so you can only evaluate anterior tooth wear. Also, your simulated incisal wear is large (0.5mm minimum) but the challenge is to detect wear much earlier than this (0.1mm). Your test models therefore have large surface changes, both on the incisal edge, and palatally (the large bulk of wire/composite).

What your paper shows is that, using 20% overlap (CC) method, you can correctly align sequential anterior teeth in the presence of significant and large shape changes (mainly the orthodontic wire). You should perhaps write the paper in these terms – related to automatically ‘ignoring’ retaining wires by using the 20% method. However, since the user still needs to manually outline the tooth, (and PC(C) and PC(D) perform with no statistical difference to CC) you might argue that the areas to ignore are obvious to the user.

Your 20% method may well fail on more subtle erosion, such as loss of 0.1mm on the occlusal surface of molars, or large palatal regions of anterior teeth losing 0.1mm. You must test this before you can advocate your method for tooth wear as it is likely that ICP will not know which 20% to use in the face of small changes (it becomes easier for the algorithm in the face of large changes)– for now, it is a method that can ignore orthodontic wire, so you should write the paper as such.

Also you should be aware there are robust versions of ICP that remove bad correspondences iteratively, without the need to specify overlap. These should be tested, as they are simpler for the user. See, for example, https://norlab.ulaval.ca/pdf/Babin2019.pdf

(These variations will certainly behave similarly to your method with the type of large differences you have in the teeth, but again, they need testing on the subtle, flat differences of erosion at 0.1mm or less)

Discussion

‘Here we present a highly accurate and informative tooth wear assessment method, which is also more convenient than the already available methods.’ This is arguably not true – there is validated freeware aimed at dentists that automatically aligns and volumetrically measures wear without needing to be an expert user. You have completely missed this development and it needs to be compared to your method before you can make such claims. (Comparing your method and the WearCompare method would be useful, and if you show your method to be simpler and more accurate, then you would have a notable contribution to the literature)

‘In the era of digitization, this is not considered a limitation, since it is expected that soon an intraoral patient scan will consist an integral part of basic dental diagnostic records.’ But you have only investigated lab scanners. We know IOS have lower resolution so cannot extrapolate your results to IOS.

‘Epidemiologic studies indicate that tooth wear, especially of the clinical crown, mainly concerns the occlusal part of the teeth due to the direct contacts with the antagonists [12,13].’ Be clear that incisal wear can be attrition, erosion, abrasion or a combination of all three. Your models only simulate attrition. Erosion is a more complex (and increasingly common) problem and the resulting change in tooth shape is different (for example cupped incisal edges with some enamel retained peripherally plus smooth but small palatal loss. Molars often show occlusal loss (again smooth, small changes over a large surface area) with the buccal and lingual aspects protected from wear.

‘Our previous study, on intact teeth in non-occlusal surfaces, suggested an estimated overlap of meshes of 30% for optimal outcomes [10].’ You are correct to point out the differences between your 2 studies. Your method risks being ‘over-fitted’ to the use-case. Really all you can conclude is that 20% works well for anterior teeth in the presence of large palatal retainers and significant wear restricted to attrition of the incisal edges. (If you were to adjust the pattern of wear, or look at premolars, you would likely find that a different % was optimal. And again for molars. And again, depending on how much tooth loss has actually occurred. Hence, rewriting you conclusion to state that, “in the presence of palatal/lingual retainers, your 20% method is capable of reliably detecting incisal attrition of 0.5mm or more, on anterior teeth” is required.

‘Thus, we expect that this technique will be also applicable on posterior teeth.’. I’m afraid I disagree. It might work, but it might not. There is no reason to confidently expect it will without trying – set up a molar, digitally erode the occlusal surface by 0.1mm (so you have a gold standard), and try your 20% CC method. It would be interesting to see.

Conclusion

This reads well except for the final sentence. It should be tested with IOS before advocating for clinical practice. Further work is required to determine whether the technique is applicable to posterior teeth, and to smaller increments of tooth wear.

Reviewer 2 Report

Review of jcm-1008551

This is an interesting contribution to tooth wear research and to the methodological development of 3D superimposition techniques. The authors base their main technical developments on their own papers from Gkandtidis et aLines  2020 and Winkler & Gkantidis 2020, however they miss the opportunity to put their findings in the broader scientific context. The manuscript main strength is the detailed methodological testing. But, I have some major doubts because I miss a validation with other available (independent) techniques and that the authors discuss the broader implications of the conducted study. Furthermore, I miss some methodological details. In addition, I would like to encourage the authors to consider developments of 3D shape and volume quantification (dental topography sensu lato) in neighbouring disciplines that could improve and help to set the results of the study in the broader context. For example, in anthropology, archaeology, biology and paleontology occlusal wear as well as changes in tooth shape and form are quantified in 3D since many years now; and occlusal wear is differentiated on µm, mm as well as cm scale, please specify on which scale the current study works and how the new technique is comparable. For a review on shape descriptors as ecometrics in dental ecology on various scales see Evans 2013 Hystrix DOI: 10.4404/hystrix-24.1-6363, relationship of dental micro- and macrowear see Schmidt 2010 AJPA DOI 10.1002/ajpa.21197, 3D topometry see Berthaume et aLines  2020 Evol Anthropol https://doi.org/10.1002/evan.21856 .

Detailed comments, suggestion, and critical points:

Line 33: “Tooth wear occurs as…” - Tooth wear is a very common and well known phenomenon with very particular reasoning in all mammals; please be more precise in the wording here and include the word “human” to be more specific; i.e. “In humans, tooth wear occurs…”.

Lines  41-51: This paragraph is about accuracy and precision, however it remains unclear what is meant by “highly accurate”, “inaccuracies”, “reduced precision”. Please help the readership to better understand what is meant and re-word; i.e. give values for the resolution in x, y, z or features / describe what the inaccuracies are.

Lines  51-66: This paragraph gives a short overview on the development of 3D scanning in clinical application in dentistry. Yet, what is missing is the clear definition on which scale such wear changes are important for clinical application. Even though the authors give a research purpose (Lines  64-66) a clear hypothesis indicating the lack of knowledge is missing and would help the reader to better understand the aim of the study.

Line  75: “dental plaster models” – please give shrinkage and resolution values of the used materials to make clear which precision is reached.

Lines  83-85: “…eighteen canines and eighteen incisors…” – please specify if upper and lower teeth are mixed and eighteen refers to number of patients.

Line  84: “manually grinded” – please refer to which grinding equipment is used (later on reference 10 is given), but for the reader it would be helpful to have the grinding equipment directly given.

Line  93: “adequate model acquisition” – unclear what is meant here, please refer clearly to the given resolution in x, y, z for the used scanner and the triangle count for the used mesh. Further, it is not clear what is meant in line 100 by “accuracy <20µm”. As a general remark, please be aware, that in neighboring scientific disciplines, there is a long history of at least 20 years of 3D scanning and measuring of tooth shape, for a review see Berthaume et aLines  2020 Evol Anthropol https://doi.org/10.1002/evan.21856; Gailer & Kaiser 2014 J Morph. doi: 10.1002/jmor.20217 ; Ackermans et al. 2020 Anat Rec DOI 10.1002/ar.24402.

Line  101: “(STL) models were further processed in Viewbox 4…” - There are two types of STL files available, please add which one is used (binary or ASCII); and specify what is meant by “further processed”.  After scanning many editing, smoothing and cropping approaches are available and some topographic measures are sensitive to smoothing and editing steps.

Line  117: “with predefined settings” – unclear what is meant, please add which settings are used.

Lines  149-152: “…characteristics of the wear volume to be measured, the slicing planes…to include the complete occlusal wear surface...edges of the crown”  versus Lines  159 “…occlusal tooth wear was defined as the difference between the two superimposed models…”.

At this point a lack of clarity in the used terms, scale and definitions becomes problematic and hinders the understanding. I would highly welcome, if the authors could clarify in more detail already in the introduction which definitions are used. What is described here sounds more like if the authors meant the 3D tooth volume loss in general and not the occlusal wear at a particular position (tooth tip or a particular occlusal area like the tooth facet). What are displayed in Figure 2 is clearly visible as buccal, lingual areas as well as interproximal areas. Please be aware, that different wear mechanisms act simultaneously or sequential and often influence each other in complex ways (for example see Pokhojaev et aLines  2018 JDR, https://doi.org/10.1177/0022034518785140); for an historical review of wear facet terminology see Koenigswald 2016 Hist Biol DOI 10.1080/08912963.2016.1256399.

Lines  185-186:  “Non-parametric statistics were applied based on abnormal distribution of the raw dta of certain variables.” – It is not clear which variables are meant; please specify which are meant.

Line  257. “material wear” – it is unclear why the authors argue about material wear at this point of the discussion. Please clarify or delete; this is highly speculative and not supported by the study, further a related hypothesis is missing.

Lines  266-269: tooth wear as a result of direct contacts with the antagonist – at this point of the discussion it is not clear why only contact with the antagonist are important for the clinical outcome and why dietary contacts are neglected. As later in the discussion Lines  286-287 and again in 294-296 the authors consider adjacent clinical crown structures to change over time, but do not give reasons. At this point of the discussion as a reader I feel lost. I highly recommend re-structuring the discussion section more clearly regarding methodological limits, duration and scale of the expected 3D changes.

Lines  306/339/340:  “true wear amount” - unclear terminology, please specify what is meant and set the value of 0.1mm3 as a value in relation to the resolution of the scanner.

Lines  321-322: other studies – there are other studies compiling 3D-quantitative tooth wear analysis in non-human mammals (see references in the text above). It would be highly interesting to compare the methodological settings as well as the outcome.

Lines  355: “…tooth types does not significantly affect outcome…” – strictly speaking this was not statistically tested. Please adjust the wording. It was just compared but a statistical model is not presented; if the authors would like to claim that statement at least a generalized linear model could be helpful to test for tooth type/position.

Round 2

Reviewer 1 Report

An improved manuscript - well done